# Unsupervised Clustering of Heartbeat Dynamics Allows for Real Time and Personalized Improvement in Cardiovascular Fitness

**DOI:** 10.3390/s22113974

**Published:** 2022-05-24

**Authors:** Cassandra Serantoni, Giovanna Zimatore, Giada Bianchetti, Alessio Abeltino, Marco De Spirito, Giuseppe Maulucci

**Affiliations:** 1Department of Neuroscience, Biophysics Sections, Università Cattolica del Sacro Cuore, Largo Francesco Vito, 1, 00168 Rome, Italy; cassandra.serantoni@unicatt.it (C.S.); giada.bianchetti@unicatt.it (G.B.); alessio.abeltino@unicatt.it (A.A.); 2Fondazione Policlinico Universitario “A. Gemelli” IRCCS, 00168 Rome, Italy; 3Department of Theoretical and Applied Sciences, eCampus University, Via Isimbardi, 10, 22060 Novedrate, Italy; giovanna.zimatore@uniecampus.it

**Keywords:** *VO_2_max*, cardiovascular fitness, machine learning, multiple modality biosignal processing, personalized medicine, physiological time series, medical technology, medical data analysis in healthcare, k-means clustering, cardiovascular risk

## Abstract

*VO*_2_*max* index has a significant impact on overall health. Its estimation through wearables notifies the user of his level of fitness but cannot provide a detailed analysis of the time intervals in which heartbeat dynamics are changed and/or fatigue is emerging. Here, we developed a multiple modality biosignal processing method to investigate running sessions to characterize in real time heartbeat dynamics in response to external energy demand. We isolated dynamic regimes whose fraction increases with the *VO_2_max* and with the emergence of neuromuscular fatigue. This analysis can be extremely valuable by providing personalized feedback about the user’s fitness level improvement that can be realized by developing personalized exercise plans aimed to target a contextual increase in the dynamic regime fraction related to *VO_2_max* increase, at the expense of the dynamic regime fraction related to the emergence of fatigue. These strategies can ultimately result in the reduction in cardiovascular risk.

## 1. Introduction

*VO_2_max* is an index of cardiovascular fitness and aerobic endurance, expressed in mL/kg·min, which refers to the maximum amount of oxygen that an individual can uptake during intense or maximum exercise [1,2,3]. It is directly proportional to the amount of energy that an individual can produce aerobically: the more oxygen consumed, the greater the energy produced [4]. Estimation of *VO_2_max* is particularly important since it was shown in several studies that cardiovascular fitness has a significant impact on overall health. Mounting evidence has firmly established that low levels of cardio-respiratory fitness are associated with a high risk of cardiovascular disease (CVD) and all-cause mortality, as well as mortality rates attributable to various cancers, especially of the breast and colon/digestive tract [5]. Those findings are supported by additional research which revealed that a 10% increase in *VO_2_max* could decrease all-cause mortality risk by 15% [6,7]. In this respect, finding a way to easily measure improvement in cardiovascular fitness is especially important to reduce cardiovascular risk. Evidence on cardiovascular risk is biased toward causes rather than prevention techniques, which have yet to be widely reproduced or supplied at a scale that makes them a viable alternative for public health efforts. Change is difficult, time consuming, and resource intensive. In this context, technology can aid in the support and maintenance of healthy behaviors: smart wearable devices can monitor and provide feedback on energy intake and expenditure [8]. To accurately measure the *VO_2_max* parameter, tests in which effort and duration put a strain on the aerobic energy system are required. The most used tests are the cycle ergometer or the treadmill in which the intensity of the exercise is gradually increased [9]. During the exercises, ventilation, and the amount of oxygen O_2_ and carbon dioxide CO_2_ contained in the inhaled and exhaled air are calculated. If oxygen consumption remains steady despite an increase in exercise intensity, *VO_2_max* is reached. However, these methods have some disadvantages: cardiopulmonary exercise test (CPET) or spirometry devices require the intervention of qualified personnel, continuous maintenance, and they are expensive. Furthermore, being bulky, they cannot be used in all environments and in everyday life [10,11]. For this reason, despite their efficiency, their use is limited to sports professionals and clinics. In recent years, wearable devices, in particular smartwatches and smart bands (i.e., Garmin Forerunner, Polar V800, Apple Watch) allow the real time and remote monitoring of physiological parameters. They are used in both medicine [12,13] and sports [14,15]. These devices will make it easier for those who wish to be healthy and change their lifestyle, as measurement and feedback systems become more refined and individualized. In particular, these devices can supply a measure of *VO_2_max* even outside the laboratory in everyday life. Some wearable devices estimate it only starting from a continuous measurement of heart rate in particular conditions (i.e., resting heart rate variability [16]) and require knowledge on gender, age, BMI, and extrapolation of the maximum heart rate frequency [17]. There are several scientific works that investigate the validity of the *VO_2_max* measurement through wearable devices, in particular wrist-worn activity trackers [18]. Kraft and Roberts [19] did not detect significant differences between the *VO_2_max* measured with spirometry and the Garmin^®^ Forerunner 920XT and reported a correlation coefficient r = 0.84 between the two signals. In the last white paper [20], Apple^®^ reported that *VO_2_max* estimation by Apple Watch is accurate and reliable relative to commonly used methods of measuring *VO_2_max*, with an average error of less than 1 MET (metabolic equivalent, with 1 MET = ~3.5 mL/kg·min) and a confidential interval (ICC) of more than 0.85. Overall, these algorithms are increasingly improving the estimates of *VO_2_max* in everyday settings, with tangible health benefits verifiable by users themselves as an increase in fitness level, benefits that are strictly correlated with increases in *VO_2_max* values [21,22,23,24].

There are several algorithms already estimating *VO_2_max* through wearable devices. *VO_2_max* is an index of entire running performance that is presented at the end of the running session. Currently, research is not able to furnish clues to understand the physiological response at the basis of *VO_2_max* improvement, a topic of extreme interest for improving athletic performance and reducing cardiovascular risk. Traditionally, knowledge of training methods to enhance endurance performance has evolved by way of trial-and-error observations of a few pioneering coaches and their athletes [25], with exercise scientists attempting to explain the underpinning biological mechanisms. Several studies have found out that interval training (IT) produces improvements in *VO_2_max* slightly greater than those typically reported with continuous training (CT) [21].

A parameter indicating in real time (i.e., during the running session) if the heart rate response is undergoing muscular, cardiovascular, and neurological adaptations underlying its improvement in response to the local external energy demand (velocity and altitude variations) would be valuable both for the user in its daily physical activity, and for clinicians to evaluate and plan the personalized training requirements. Indeed, this response may change from person to person according to genetics, diet, and type and quantity of individual activity patterns. Within this framework, in this work we introduce a multimodal analysis method that indicates if the heart rate response is experiencing the muscular, cardiovascular, and neurological adaptations that underpin its improvement in response to the local external energy demand in real time (during the sport session) (velocity and altitude variations). These physiological responses are at the basis of *VO_2_max* improvement. A k-means clustering algorithm was trained on heart rate, velocity, and altitude features to classify time series intervals of the running sessions. We show that it is possible to isolate four intervals characterized by peculiar heart rate dynamics. Among these, a dynamic range is characterized by a fraction which highly correlates with the *VO_2_max* parameter, and another dynamic range presents some features that can be associated with the emergence of fatigue. This ultimately allows users to understand when, and to what extent, cardiovascular response is adapting to improve *VO_2_max*; thus, providing personalized real-time feedback about the user’s fitness level improvement.

## 2. Materials and Methods

### 2.1. Data Acquisition

This study considered city running sessions of comparable duration performed by a male non-professional runner over a year (age = 57 years; BMI = (22.63 ± 1.85) kg/m^2^) who has shown an improvement in his *VO_2_max* value (35.94 ± 1.91 mL/kg·min). The runner acquired 21 different time series (duration 1 h) using an Apple Watch (A2292) and, for a subset of the acquisitions the same data were acquired using a smart watch (Garmin Fenix 5x Plus), respectively one on the right wrist and one on the left wrist. Apple Watch provides heart rate (*HR*) time series (measured in beats per minute, bpm), speed (*v*) time series (measured in meters per second, m/s), and altitude (*z*) time series (measured in meters, m) at non equally spaced time intervals (respectively, (5.1 ± 2.6) s, (2.6 ± 1.4) s, (2.0 ± 1.8) s). Garmin provides heart rate time series (measured in beats per minute, *HR*) at equally spaced time intervals (*t* = 1 s). The reason why we chose to use both sensors is that an equally spaced signal allowed us to calculate the *HR* auto-correlation function (ACF), without any imputing and manipulation on raw data, needed to select the optimal time window to split up the time series and extract the features for the clustering analysis. Notice that we obtained a Pearson correlation coefficient r = 0.93 between the time series of apple and that of Garmin, meaning that they are extremely similar (Appendix A).

### 2.2. Data Cleaning

For this analysis we focused on the interval including *HR* values which are related to intense and maximum physical activity (area of maximum cardiac effort). This interval will include, therefore, only *HR* values above 90% of the maximum heart rate, calculated according to Tanaka formula [26]:(1)$$HR_{max}=208-0.7\times age$$

*HR* time series were further processed by removing outliers due to periodic sensor malfunctions using z-score with a threshold of 3 standard deviations. Speed time series, affected by the highest signal to noise ratio due to GPS, were processed with a low pass Butterworth filter with a cutoff frequency of 0.01 and order 2. The values of these parameters were chosen following a noise signal analysis. After data cleaning the different time series were time aligned. As every running session was performed at different altitudes, we rescaled the altitude timeseries subtracting the relative starting point *z*_0_.

### 2.3. Model

#### 2.3.1. Time Interval as Statistical Unit

We divided each time series in *n* overlapping point by point time intervals of width Δ*t* = 90 s, with *n* being the total number of points in the time series. Δ*t* was selected by calculating the ACF cutoff time (*t_cut_*) for the *HR* time series. ACF defines how data points in a time series are related, on average, to the preceding data points [27]. The red point is the intersection between ACF and the upper border of the confidence interval. The red point thus indicates a threshold lag *t_cut_*, suggesting that at times higher than *t_cut_* a correlation can be found with a probability less than 5%.

Figure 1 shows an example of the ACF function for a single running session. We can see that *HR* values are correlated with lag times until *t_cut_* ≃ 80 s. This value is the one corresponding to the point in which the ACF function cuts the upper confidence threshold (red point in Figure 1) [28].

Since raw *HR* time series data were not stationary, we performed a detrending using the polyfit function in the numpy package [30]. After performing the Dickey–Fuller test for stationarity [31], we extracted the intersection value between ACF and upper confidence interval for the subgroup of running sessions acquired with Garmin since, as already mentioned in the Data Acquisition paragraph, Garmin provides heart rate *HR* time series at equally spaced time intervals (Δ*t* = 1 s). We selected 16 running sessions and we performed a statistical analysis on the intersection values of the ACF with confidence intervals, which gave us a time window of width (90 ± 26) s. ACF cutoff lags (*τ_ACF_*) are not correlated with physical fitness (Appendix A).

#### 2.3.2. Feature Selection for the Clustering Algorithm

For each interval with width *t**_cut_* = 90 s, we selected two features considering *HR*, speed *v*, and altitude *z* according to the following criteria. These features are two terms accounting for *HR* variation (Δ*HR*) and external energy demand variation (Δ*E*). The external energy demand is described by the term *E,* resembling mechanical energy: it includes a potential energy term ($V=g(z-z_{0}$)) and a kinetic energy term ($K=\frac{1}{2}v^{2}$). The two energy terms are normalized separately in a range of [0, 1]. In this way, both terms have comparable weights in the total energy, which can then vary in a range [0, 2].

Our aim is to cluster time intervals with an unsupervised method to classify them according to the relationship between external energy demand variation and heart rate variation. To express the variation in the signal and simultaneously guarantee optimal clustering performances we defined the following features:(2)$$\Delta E=\gamma_{0}\left(E\right)+\gamma_{1}\left(E\right),\;\;\;\;\Delta HR=\gamma_{0}\left(HR\right)+\gamma_{1}\left(HR\right)$$
where $\gamma_{0}\left(x\right)=\frac{\left(\;x\;¯-x_{0}\right)}{\sigma },\gamma_{1}\left(x\right)=\frac{\sum_{i=1}^{n}{\left(x_{i}-\;x\;¯\right)}^{3}}{\sigma^{3}}$ is the skewness index and it is a measure of the asymmetry of the probability distribution of a real-valued random variable about its average. In these expressions, σ is the standard deviation, $\underline{x}$ is the mean value of the feature in the time interval, and *x*_0_ is the initial point of the time interval. We used these features since variations are subjected to noise and are not always linear. The first term expresses the position of the mean of the points in the considered time interval with respect to the starting point of the interval, and the second is a reinforcing term expressing the skewness of the distribution. We provide below evidence that these features are able to discriminate variations in the signals on noisy and non-linear data (paragraph Section 3.1). Note that Δ*E* and Δ*HR* are mass independent since they are normalized to their standard deviation.

#### 2.3.3. Clustering Analysis Algorithm

Once the values of the clustering features chosen were calculated for each interval of width Δ*t*, cluster analysis was performed.

Clustering analysis is an unsupervised statistical method for processing data consisting of the organization of data into groups called clusters, which has many applications as market research [32], pattern recognition [33], data analysis [34], and image processing [35,36]. Clustering is measured using intracluster distance, the distance between the data points inside the cluster, and intercluster distance, the distance between data points in different clusters. A good clustering analysis minimizes the intracluster distance meaning that a cluster is more homogeneous and maximizes the intercluster distance. We chose the *k*-means algorithm [37,38] which tries iteratively to partition the dataset into *K* predefined distinct non-overlapping subgroups. *k*-means represents each of the *k* clusters *C_j_* by the mean (or weighted average) *c_j_* of its points (centroid). The sum of distances between elements of a set of points and its centroid expressed through an appropriate distance function is used as the objective function. We employed the L2 norm-based objective function, i.e., the sum of the squares of errors between the points and the corresponding centroids, which is equal to the total intracluster variance:(3)$$E\left(C\right)=\overset{k}{\underset{j=1}{\sum }}\overset{\;}{\underset{x_{j}\in C_{j}}{\sum }}{\left|\left|x_{j}-C_{j}\right|\right|}^{2}.$$

The *k*-means algorithm can be summarized in four main steps:Specify the number of clusters *k* and initialize centroids $C=\left\{c_{1},c_{2}\ldots c_{k}\right\}$ by randomly selecting K data points for the centroids without replacement;For each j∈{1, . . . ,k}, set the cluster *C_j_* to be the set of points in X that are closer to c_j_ than they are to *c_j_* for all *i* ≠ *j*;For each j∈{1, . . . ,k}, set *c_i_* to be the center of mass of all points in *C_j_*: $c_{j}=\frac{1}{\left|C_{i}\right|}\overset{\;}{\underset{x\in C_{i}}{\sum }}x$;Repeat steps 2 and 3 until a stopping criterion is achieved (no reassignments with tolerance < 10^−5^).

This version, known as Forgy’s algorithm [38], works with any Lp norm and it does not depend on data ordering. A weakness of the *k*-means algorithm is that after a certain time it will always converge due to a local minimum, and this is strictly connected to the starting centroid choice. One method to help address this issue is the *k*-means ++ scheme [39,40] which initializes the centroids to be distant from each other, leading to probably better results than random initialization.

Unsupervised clustering analysis was performed with Python 3.8.5 [41] and scikit-learn 1.0.2 package [42].

#### 2.3.4. Choosing the Best *k* Number

The optimal number of clusters was determined using the silhouette method [43].

The silhouette score is a very useful index for the quality of clustering analysis. It is a measure of how similar an object is to its own cluster (cohesion) compared to other clusters (separation). Given a cluster *A* and any object *i* in the data set, when cluster *A* contains other objects except *i*, we can compute the distance of object *i* with all other points in the same cluster: (4)$$a_{i}=\frac{1}{n_{A}-1}\overset{\;}{\underset{j\in A,j\neq i}{\sum }}d_{ij},$$
where *n**_A_* is the number of points belonging to cluster *A* and *d**_ij_*is the mean distance between data points *i* and *j* and in the cluster *A*.

Considering any cluster *B* which is different from *A*, we can compute: (5)$$b_{i}=\frac{1}{n_{B}}\underset{j\in B}{\sum }d_{ij}\;$$
which gives the smallest mean distance of *i* to all points in any other cluster of which *i* is not a member. With these two distances, we can define the silhouette value for the point *i:*(6)$$s_{i}=\frac{b_{i}-a_{i}}{\left\{a_{i},b_{i}\right\}\;}$$
which is set to be 0 when cluster *A* contains a single object. For every object *i*, the silhouette value can vary in the range [−1, +1]. A silhouette value near +1 indicates that point *i* is far away from the neighboring clusters. A value of 0 indicates that point *i* is on or very close to the decision boundary between two neighboring clusters and negative values indicate that point *i* might have been assigned to the wrong cluster.

The average of all silhouette values for each point *i* in the dataset returns the silhouette score:(7)$$\bar{S}=\sum_{i=1}^{N}s_{i}.$$

### 2.4. Recurrence Quantification Analysis (RQA)

Time series were analyzed in terms of Recurrence Quantification Analysis (RQA) which can be defined as a graphical, statistical, and analytical tool [44,45] used by several disciplines from physiology [46,47,48,49,50,51] to earth science [52,53,54] and economics [55,56,57]. The RQA-based method employed in the analysis of *HR* time series is widely explained in earlier papers [49,50]. To perform RQA computation we used a software written in Python [58]. The RQA input values used are embedding = 7; lag = 1; radius = 5; line = 4; Euclidean distance.

### 2.5. Statistics

Differences among clusters were determined by conducting an ANOVA and Kruskal–Wallis test for the data that were not normally distributed. Normal data distribution was assessed by visual inspection, variance comparison and Shapiro–Wilk’s test. Subsequently, we performed a Mann–Whitney–Wilcoxon post-hoc test for non-normal distributions and Tukey post-hoc test for normal distributions both with Bonferroni adjustment for *p*-values. Values of *p* < 0.05 were considered statistically significant.

## 3. Results

### 3.1. K-Means Clustering Reveals Four Dynamic Clusters

The aim of our clustering strategy was to find different dynamical regimes during running sessions using different metabolic processes. Clustering parameters are rescaled using as an offset the mean value of the distribution, and as scaling factor α_s_, where s is the standard deviation of the distribution and α a tunable parameter. If the distributions are almost Gaussians, α > 3 ensures that more than 99% of the values are considered. This was verified by visual inspection and qq-plots. When there are significant deviations from Gaussians the α factor is adjusted to include at least 99% of values by direct calculation. The result is a *k*-means classification with four clusters, that we will call dynamic clusters, as shown in Figure 2.

The silhouette plot in Figure 2a displays a local maximum at the chosen *k* = 4 (highlighted by the red dashed line) and has an average silhouette value of 0.54.

Figure 2b shows the clustered data. For simplicity, we indicate the clusters with the respective signs of the variations: +/+ cluster (yellow), −/− cluster (black), −/+ cluster (blue), and +/− cluster (green). In clusters +/+ (yellow) and −/− (black), Δ*HR* and Δ*E* are directly proportional. The +/+ cluster is characterized by positive Δ*HR* (mean ± sd = 1.16 ± 0.52) and positive Δ*E* (mean ± sd = 1.24 ± 0.46). The −/− cluster is characterized by negative Δ*HR* (mean ± sd = −1.20 ± 0.58) and negative Δ*E* (mean ± sd = −1.25 ± 0.52). Contrariwise, −/+ (blue) clusters and +/− (green) are linked to inversely proportional regions for Δ*HR* and Δ*E*. The −/+ cluster is characterized by positive Δ*HR* (mean ± sd = 1.13 ± 0.55) and negative Δ*E* (mean ± sd = −1.29 ± 0.40). The +/− cluster is characterized by negative Δ*HR* (mean ± sd = −1.06 ± 0.60) and positive Δ*E* (mean ± sd = 1.30 ± 0.32). We can observe these results in Figure 3a,b. In Figure 3c representative clustered time intervals from a single running session are shown. We can observe that clustering identifies coherently the areas of growth and decreases in these quantities: when Δ*HR* and Δ*E* are both positive, *HR(t)* and *E(t)* are increasing in the considered time interval; when Δ*HR* and Δ*E* are both negative, *HR(t)* and *E(t)* are decreasing in the considered time interval. When they have different signs, the positive signal increases whilst the negative decreases. To show that this is the case, we performed a linear regression analysis on the same time windows of the clustering analysis. The signs of the slopes are in good agreement with the signs of the variations in the features identified in those specific time windows, except for a small fraction (between 10% and 20%) in which strong non-linearity affects the goodness of linear fits (Appendix A). This happens when Δ*E* and Δ*HR* are very close to zero and/or when *HR(t)* and *E(t)* undergo both increases and decreases in the same time window. However, features reported in Equation (2) can capture the general tendency of these unfitted data to increase or decrease: while $\gamma_{0}$ is in these cases characterized by a very small value, the skewness $\gamma_{1}$ is still relevant and, with exception of perfectly symmetrical data distributions, the sign of the variation takes into account the general tendency of the data to be above or below the average value (which is almost zero). Appendix A shows some representative time windows in which the clustering analysis effectively identifies an increase or decrease in the features.

### 3.2. Descriptions of the Dynamic Clusters

Table 1 shows general characteristics of statistical quantities in different clusters. We chose a subgroup of clusters relative to non-overlapping windows to perform a statistical analysis on independent points. Clusters do not differ for average *HR*, average speed, *HR* standard deviation, speed standard deviation, and altitude standard deviation. The only significant quantities are then the clustering quantities.

### 3.3. Temporal Mapping of Clusters on Running Sessions

Figure 4 shows cluster frequencies in different running sessions. Bar plots highlight an upward trend in the number of percentage occupancy of cluster −/+ (blue) and a downward trend for cluster −/− (black) in four different running sessions acquired three months apart from each other. The trends agree with the increasing value of the *VO_2_max* parameter over time reported. Values of *VO_2_max* and percentage occupation number of clusters −/+ and −/− in each running session are reported in Table 2.

### 3.4. Fraction of Cluster −/+ Is Positively Correlated with VO_2_max, While Fraction of Cluster −/− Is Negatively Correlated

In addition to what has been said in the previous paragraph, we can observe in Figure 5 the correlations between the cluster frequencies for each cluster in the different running sessions and the *VO_2_max* value estimated with the Apple Watch in the same sessions. Our analysis found a positive correlation between *VO_2_max* and cluster −/+ (r = 0.72, Figure 5b) and a negative correlation between *VO_2_max* and cluster −/− (r = −0.52, Figure 5a), though the r value is slightly less. Projections of the clusters on representative HR, speed, and altitude time series are shown in Figure 6a: as an example, a running session acquired in June 2021 is reported. The distribution of the clusters along the curve seems isotropic.

### 3.5. Temporal Distribution of the Heartbeat Dynamics and Correlation with Neuromuscular Fatigue

To deeply investigate the temporal distribution, we divided the time series into three equal sections and calculated the percentage concentrations of the clusters in the individual sections. To compare even slightly different runs in overall duration we normalized time between 0 (start time) and 1 (end time). We called the three sections “start” (normalized time interval 0–0.33), “middle” (normalized time interval 0.33–0.66), and “end” (normalized time interval 0.66–1). Characteristic trends can be seen for the various clusters over time. Results of this analysis are shown in Figure 6b–e.

Performing a repeated measure ANOVA with a Tukey HSD post-hoc comparison, we can observe that the cluster frequencies of the −/− cluster in Figure 6b in the “start” section is significantly higher than the “middle” section. A decreasing trend can be observed for cluster −/+ in Figure 6c. The cluster frequencies in every section are all significantly different.

In Figure 6d,e, frequencies for both +/− and +/+ clusters present an increasing trend from the “start” section to the “end” section: cluster frequencies in the “end” section are significantly higher than “start” and “middle” section. Values of the cluster frequencies experience a saturation. We performed RQA analysis on the same sections of the analysis just presented. An important parameter of RQA analysis is determinism (DET) which is an indicator of the regularity or complexity of the system dynamics. In earlier works, a correlation between high values of DET and fatigue during submaximal incremental exercise has been found [49,50]. We found an increasing trend for DET in every running session. Repeated measures ANOVA and post-hoc Tukey with Bonferroni corrections revealed significant differences between the “end”-“middle” section and “end”-“start” section. Figure 7a reports boxplots of determinism in the three different sections. In Figure 7b–d an example of a recurrence plot in the three different sections is shown. The number of black points (called recurrence points) from the “start” section to “end” section increases as the value of the determinism (values of ANOVA and Tukey tests are reported in Appendix A).

## 4. Discussion and Conclusions

Several studies have found out that interval training (IT) produces improvements in *VO_2_max* slightly greater than those typically reported with continuous training (CT) [21]. These training approaches are indeed aimed to abruptly change the external energy demand ΔE. In this respect, if during city running abrupt variations in velocity or height due to the alternation of uphill and downhill slopes with flat areas occur, an improvement in the cardiovascular systems in copying with these stresses can trigger heart rate dynamics related to a *VO_2_max* increase. To investigate this point, in this manuscript we considered city running sessions of comparable duration performed by a non-professional runner over a year. We calculated Δ*E* and Δ*HR* features from the raw data extracted from the Apple Watch in overlapping point by point time intervals of width Δ*t* = 90 s, because of the statistical analysis on the intersection values of the ACF with confidence intervals on 16 different running sessions (Figure 1). The heart rate ACF was decisive for determining the time window in which heart rate values were still related to previous events. The clustering analysis identified four different cluster dynamics of heartbeat in response to external energy demand (+/+, +/−, −/+, −/−). In directly proportional clusters (+/+ and −/−) an increase (decrease) in the external demand is correlated to an increase (decrease) in the cardiovascular response to adapting to the cardiac output needed to fulfill increased oxygen requirements. While these dynamics of the cardiovascular response are standard and traceable also on longer timeframes, we found two peculiar dynamics grouped in −/+ and +/− clusters which are not characterized by this straightforward relation. In the −/+ cluster, Δ*HR* increases despite Δ*E* decreasing, i.e., cardiovascular response increases despite the energetic demand of the environment decreasing (i.e., z and/or v decrease). We observed a positive correlation with *VO_2_max* and the frequency of the −/+ cluster (r = 0.72, Figure 5a). This increase is at the expense of the −/− cluster whose frequency inversely correlates with *VO_2_max* (r = −0.52, Figure 5b). Moreover, investigation of the temporal distribution of the clusters by classifying running sessions in three equal time intervals (“start”, “middle”, and “end”) shows that the −/+ cluster percentage decreases from the “start” to the “end”. Since the increase in DET measured through RQA analysis (Figure 7), as described in previous publications, indicates the appearance of neuromuscular fatigue [49,50], we hypothesize that −/+ dynamics are especially active when the organism is not experiencing fatigue. Overall, we can hypothesize that the physiological processes connected to a *VO_2_max* increase, such improvement in cellular metabolism of muscular cells in the mechanisms characterizing oxygen uptake [59], and vascular modifications leading to a better oxygen distribution, could change cellular responses to stimuli ultimately leading to a hyper-stimulation of the sympathetic nervous system. This altered response lasts until the emergence of neuromuscular fatigue occurs. This neuromuscular modification leading to a new regime of sympathetic stimulation can therefore be connected with the increase in frequency of +/− cluster dynamics. The dynamics of the +/− cluster, in which Δ*HR* decreases despite Δ*E* increasing, does not present a significant correlation with *VO_2_max* variation, indicating that it is not related to the physiological improvement. This cluster may instead be related to a delayed physiological response to an increase in the external energy request and fatigue. The +/− fraction increase from “start” to “end” can be therefore related to the fact that a delayed cardiovascular response increases with fatigue.

Overall, the combination of these results can be extremely valuable in providing personalized exercise plans. Indeed, since it is possible to detect characteristic heartbeat dynamics, the possibility to provide personalized feedback about the user’s fitness level improvement is opened: improvements in cardiovascular fitness may be realized developing personalized exercise plans aimed at targeting a contextual increase in the −/+ fraction, related to *VO_2_max* increase, at the expense of the +/− fraction, related to the emergence of fatigue. These strategies can ultimately result in the reduction in cardiovascular risk and in the risk of developing other devastating pathologies such as cancer. This study, by presenting a new method of analysis, is limited to a single subject, as the analysis is conceived as person-centered by extracting features that would be hidden by the variability between individuals. However, this innovative analysis is widely applicable and has implications beyond the specific case. Other subjects, analyzed with the same method, could display similar or differing features according to their medical history, age, sex, and fitness status. Further research will generalize these results to improve the extraction of cardiovascular fitness improvement features from wearable devices and the physiological interpretations of the signals belonging to each cluster.

## Figures and Tables

**Figure 1 sensors-22-03974-f001:**
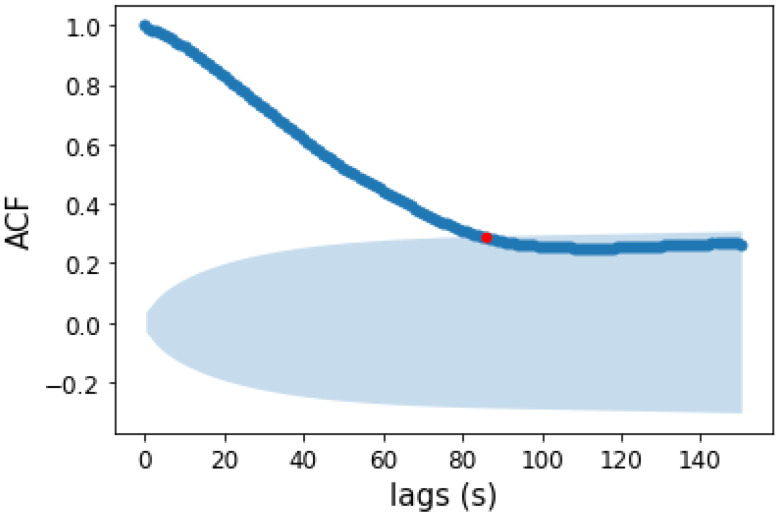
ACF plot of *HR* for a single running session. The blue shaded region is the confidence interval with a value of α = 0.05 calculated with Bartlett’s formula [29]. Anything within this range represents a value that has no significant correlation with the most recent value for the *HR*. In this case, we observe significant correlations from 0 to 80 s. The red point is the intersection between the ACF function and the upper confidence threshold. Correlations subsequent to that point are no longer significant.

**Figure 2 sensors-22-03974-f002:**
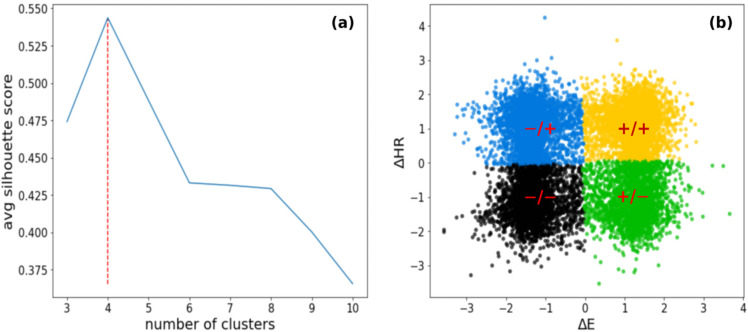
Unsupervised *k*-means clustering method. (**a**) Number of clusters vs. average silhouette score for *k*-means clustering on sample data. The red dashed line indicates the maximum average score for *k* = 4 clusters. (**b**) Visualization of clustered data for *k* = 4 in the Δ*E* vs. Δ*HR* plane. The clusters have been named with the respective signs of Δ*E* and Δ*HR* variations: +/+ cluster (yellow), −/− cluster (black), −/+ cluster (blue), and +/− cluster (green). The −/+, +/+, +/−, and −/− in the figure represent the centroids of the corresponding cluster.

**Figure 3 sensors-22-03974-f003:**
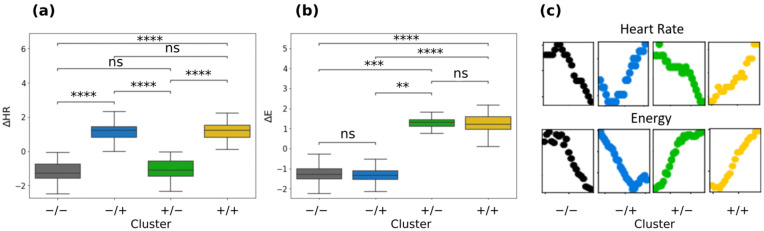
(**a**) Δ*HR* boxplots for every cluster; (**b**) Δ*E* boxplots for every cluster; (**c**) heart rate (up) and energy (down) in running fragments belonging to different clusters. The clustering analysis assigns the belonging of these fragments to different clusters coherently with the sign of both Δ*HR* and Δ*E*. *p*-value annotation legend: ns: 5.00 × 10^−2^ < *p* ≤ 1.00 × 10^0^; **: 1.00 × 10^−3^ < *p* ≤ 1.00 × 10^−2^; ***: 1.00 × 10^−4^ < *p* ≤ 1.00 × 10^−3^; ****: *p* ≤ 1.00 × 10^−4^. Color legend: yellow: +/+ cluster; black: −/− cluster; blue: −/+ cluster; green: +/− (see Figure 2 in Section 3.1).

**Figure 4 sensors-22-03974-f004:**
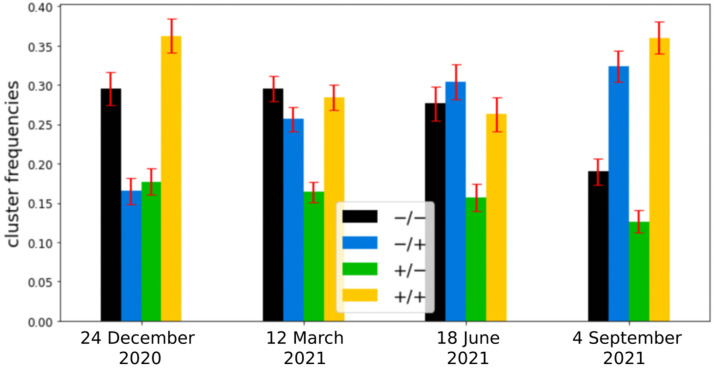
Bar plots with bar errors (red) of cluster frequencies in four different running sessions acquired three months apart from each other. −/+ cluster (blue) and −/− cluster (black) cluster frequencies seem, respectively, to increase and decrease with time.

**Figure 5 sensors-22-03974-f005:**
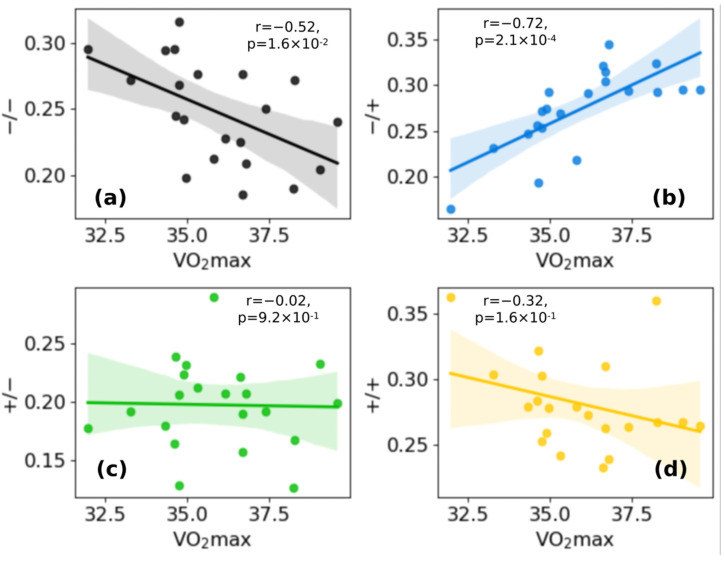
*VO_2_max* vs. cluster frequencies for every cluster: (**a**) cluster −/−; (**b**) cluster −/+; (**c**) cluster +/−; (**d**) cluster +/+. A positive correlation between *VO_2_max* and cluster −/+ and a negative correlation between VO_2_max and cluster +/+ can be observed. Legend: Pearson correlation coefficient (r), *p*-value (*p*). Color legend: yellow: +/+ cluster; black: −/− cluster; blue: −/+ cluster; green: +/− (see Figure 2 in Section 3.1).

**Figure 6 sensors-22-03974-f006:**
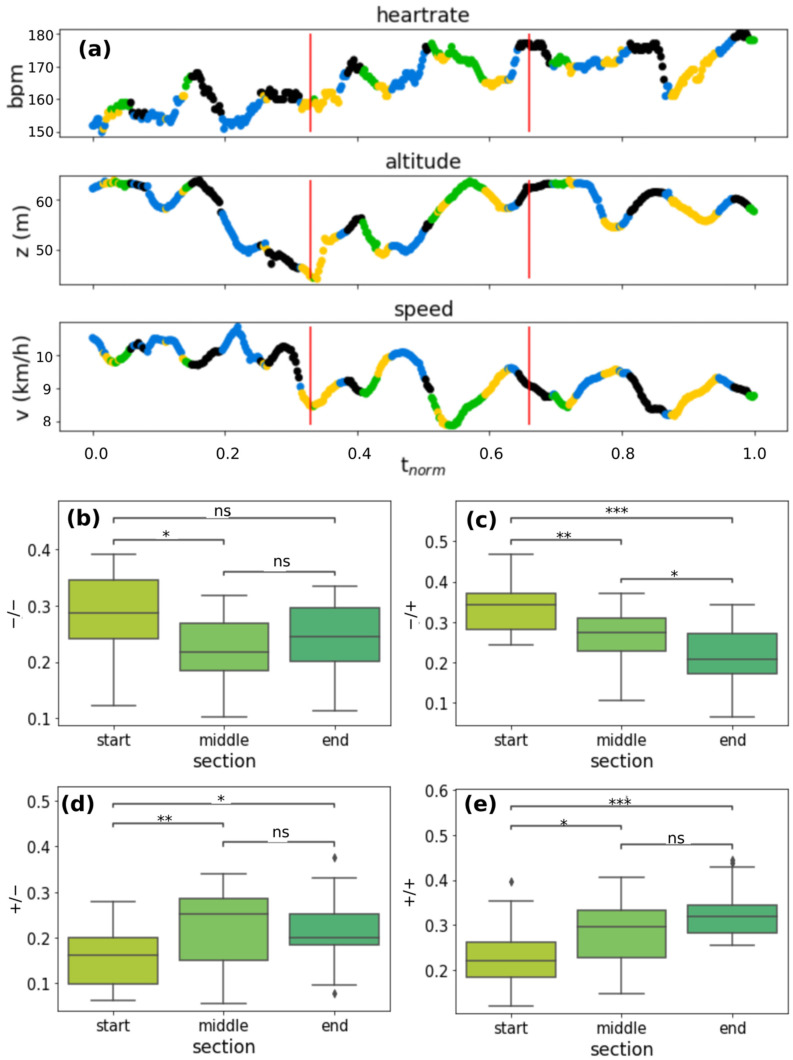
(**a**) Projection of the clusters along *HR*, speed, and altitude time series related to a running session acquired in June 2021. The normalized time between 0 and 1 is shown on the *x* axis. The vertical red lines divide the time series into the three sections “start” (from 0 to 0.33), “middle” (from 0.33 to 0.66), and “end” (from 0.66 to 1). Color legend: yellow: +/+ cluster; black: −/− cluster; blue: −/+ cluster; green: +/− (see Figure 2 in Section 3.1.); cluster frequency boxplots in the three different running sections for cluster −/− (**b**), cluster −/+ (**c**), cluster +/− (**d**), and cluster +/+ (**e**). *p*-value annotation legend: ns: 5.00 × 10^−2^ < *p* ≤ 1.00 × 10^0^; *:1.00 × 10^−2^ < *p* ≤ 5.00 × 10^−2^; **: 1.00 × 10^−3^ < *p* ≤ 1.00 × 10^−2^; ***: 1.00 × 10^−4^ < *p* ≤ 1.00 × 10^−3^.

**Figure 7 sensors-22-03974-f007:**
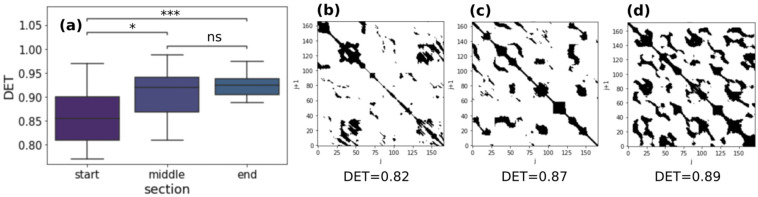
(**a**) RQA determinism boxplots in three different running sections; a representative RP of *HR* time series during an exemplary running session. In “middle” (**b**) and “end” (**c**) sections a larger number of black points (recurrence points) can be seen with respect to the “start” section (**d**). *p*-value annotation legend: ns: 5.00 × 10^−2^ < *p* ≤ 1.00 × 10^0^; *: 1.00 × 10^−2^ < *p* ≤ 5.00 × 10^−2^; ***: 1.00 × 10^−4^ < *p* ≤ 1.00 × 10^−3^.

**Table 1 sensors-22-03974-t001:** General characteristics of the clustered population. *p*-value annotation legend: ns: 5.00 × 10^−2^ < *p* ≤ 1.00 × 10^0^; **: 1.00 × 10^−3^ < *p* ≤ 1.00 × 10^−2^; ***: 1.00 × 10^−4^ < *p* ≤ 1.00 × 10^−3^; ****: *p* ≤ 1.00 × 10^−4^.

	Clusters					Post-Hoc Comparison					
Features	Black +/+ (*n* = 172) ^1^	Blue −/+ (*n* = 179) ^1^	Green +/− (*n* = 131) ^1^	Yellow −/− (*n* = 149) ^1^	*p*-Value^2^ (Kruskall)	+/+ vs.			−/+ vs.		+/− vs.
						−/+	+/−	−/−	+/−	−/−	−/−
HR mean (bpm)	162.44 ± 8.33	163.33 ± 8.64	163.97 ± 8.89	161.95 ± 9.89	0.47						
V mean (km/h)	9.24 ± 0.81	9.21 ± 0.77	9.01 ± 0.68	9.05 ± 0.79	0.21						
HR St. Dev. (bpm)	1.83 ± 1.05	1.99 ± 1.30	1.54 ± 0.72	2.14 ± 1.30	0.06						
V St. Dev. (km/h)	0.17 ± 0.18	0.21 ± 0.14	0.19 ± 0.15	0.16 ± 0.12	0.16						
Z St. Dev. (m)	1.22 ± 0.85	1.10 ± 0.79	1.26 ± 0.97	1.32 ± 0.88	0.43						
∆*E*	1.24 ± 0.46	−1.29 ± 0.40	1.30 ± 0.32	−1.25 ± 0.52	<0.0001 (****)	<0.0001 (****)	ns	<0.0001 (****)	0.004 (**)	ns	<0.001 (***)
Δ*HR*	1.16 ± 0.52	1.13 ± 0.55	−1.06 ± 0.60	−1.20 ± 0.58	<0.0001 (****)	ns	<0.0001 (****)	<0.0001 (****)	<0.0001 (****)	<0.0001 (****)	ns

^1^ Mean ± SD or frequency (%); ^2^ Fisher’s exact test; Kruskal–Wallis rank sum test.

**Table 2 sensors-22-03974-t002:** Cluster +/− and −/− cluster frequency and *VO_2_max*.

Date	24 December 2020	12 March 2021	18 June 2021	4 September 2021
*VO_2_max* (mL/kg·min)	31.96	34.59	36.7	38.23
% Cluster −/+ (blue)	0.16 ± 0.02	0.26 ± 0.02	0.30 ± 0.02	0.32 ± 0.02
% Cluster −/− (black)	0.29 ± 0.02	0.29 ± 0.02	0.27 ± 0.02	0.19 ± 0.02

## Data Availability

The data presented in this study are available on request from the corresponding author.

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
