# Peer review of "Unsupervised Clustering of Heartbeat Dynamics Allows for Real Time and Personalized Improvement in Cardiovascular Fitness"

_sensors, 2022, doi:10.3390/s22113974_

Round 1
Reviewer 1 Report
In '2.3.1. Clustering analysis algorithm' the authors should describe the algorithm (steps).
In figure 1, what represents the redpoint?
In Figure 2 which graph is b and a? In the second graph what represents the -/+, +/+, +/-, -/-?
Author Response
"Please see the attachment."

Reviewer 2 Report
The title is too long and the contribution is unclear. Please consider improving the way you use to deliver information:
- Why did you decide to investigate this topic?
- What is the core contribution of your work?
- What makes it innovative and worth a journal publication? In other words, what problems did you solve, but the other research teams could not resolve?
The IRB approval seems very questionable. Why could you use a 2017 approval on a 2022 paper submission? It has been over 5 years.
Author Response
"Please see the attachment."

This manuscript is a resubmission of an earlier submission. The following is a list of the peer review reports and author responses from that submission.